# Insights into the Potential Mechanisms of JAK2V617F Somatic Mutation Contributing Distinct Phenotypes in Myeloproliferative Neoplasms

**DOI:** 10.3390/ijms23031013

**Published:** 2022-01-18

**Authors:** Panhong Gou, Wenchao Zhang, Stephane Giraudier

**Affiliations:** 1Laboratoire UMRS-1131, Ecole doctorale 561, Université de Paris, 75010 Paris, France; 2INSERM UMR-S1131, Hôpital Saint-Louis, 75010 Paris, France; 3BFA, UMR 8251, CNRS, Université de Paris, 75013 Paris, France; zhangwch611@gmail.com; 4Service de Biologie Cellulaire, Hôpital Saint-Louis, AP-HP, 75010 Paris, France

**Keywords:** MPN, JAK2V617F

## Abstract

Myeloproliferative neoplasms (MPN) are a group of blood cancers in which the bone marrow (BM) produces an overabundance of erythrocyte, white blood cells, or platelets. Philadelphia chromosome-negative MPN has three subtypes, including polycythemia vera (PV), essential thrombocythemia (ET), and primary myelofibrosis (PMF). The over proliferation of blood cells is often associated with somatic mutations, such as *JAK2*, *CALR*, and *MPL*. JAK2V617F is present in 95% of PV and 50–60% of ET and PMF. Based on current molecular dynamics simulations of full JAK2 and the crystal structure of individual domains, it suggests that JAK2 maintains basal activity through self-inhibition, whereas other domains and linkers directly/indirectly enhance this self-inhibited state. Nevertheless, the JAK2V617F mutation is not the only determinant of MPN phenotype, as many normal individuals carry the JAK2V617F mutation without a disease phenotype. Here we review the major MPN phenotypes, JAK-STAT pathways, and mechanisms of development based on structural biology, while also describing the impact of other contributing factors such as gene mutation allele burden, JAK-STAT-related signaling pathways, epigenetic modifications, immune responses, and lifestyle on different MPN phenotypes. The cross-linking of these elements constitutes a complex network of interactions and generates differences in individual and cellular contexts that determine the phenotypic development of MPN.

## 1. Introduction

### 1.1. Myeloproliferative Neoplasms

The myeloproliferative neoplasms (MPN) are a heterogeneous group of disorders characterized by the implication of one or more of the myeloid lineages. The myeloproliferative disease was first defined by the hematologist William Dameshek in 1951 [1] and reclassified as blood cancer in 2008 by the World Health Organization (WHO) [2]. MPN arises in the hematopoietic stem cell (HSC) compartment due to a single HSC acquiring somatic mutations (the most frequent mutation in MPN is JAK2V617F) that give the mutant HSC a selective advantage over normal HSC, thereby promoting myeloid cell differentiation and producing a myeloproliferative phenotype. In 2016, WHO classified MPN into four categories, based on the Philadelphia chromosome. The Philadelphia chromosome-positive includes chronic myeloid leukemia (CML) as well as in a subset of acute lymphoblastic leukemia (ALL), and the Philadelphia chromosome-negative has three subtypes, including polycythemia vera (PV), essential thrombocythemia (ET), and primary myelofibrosis (PMF). Of these, the Philadelphia chromosome-negative ones are closely related to JAK2V617F. 

Classical PV presentation comprises red blood cell mass increase, hematocrit (HCT) increase, and leukocyte count increase, frequently associated with splenomegaly and incidental myelofibrosis. The HCT of peripheral blood is not always representative of the true volume of red cells in PV patients. Plasma volume can also vary; genuine erythrocytosis is indistinguishable from pseudo-erythrocytosis related to plasma volume contraction. Diagnosis is usually reached when there is splenomegaly, leukocytosis, and thrombocytosis. The JAK2V617F mutation is present in at least 95% of PV patients [3,4,5]. If PV is suspected, but there is no JAK2V617F mutation, mutations in exon 12 of JAK2 should be searched. Clinically, a characteristic of PV is a high frequency of thrombotic events concomitant to MPN, usually happening at the early course of the disease. PV is an unstable disease and may evolve toward myelofibrosis or leukemia. Treatment consists of venesections, anti-proliferative therapy, or JAK2-targeting drugs. However, cytoreductive treatments like pipobroman or hydroxyurea not only do not eradicate the disease but have also been suggested to increase the progression rate to AML [6]. Ruxolitinib, a JAK2 inhibitor, has been recognized to induce hematological responses, but the reduction of the clonal development [7] due to this drug is not established. Interferon-alpha (IFN-α) seems to be the safest treatment available to control leukocytosis, thrombocytosis, and extramedullary hematopoiesis and to control symptoms caused by inflammatory cytokines. It is also able to reduce the development of the clonal population [8].

ET primarily occurs in people older than 60 years. However, it can also be diagnosed in young people, especially females, with approximately 20% of ET diagnoses in people younger than 40 years [9,10,11]. ET patients present with thrombocytosis, develop persistent and progressive thrombocytosis, and, as in PV, arterial and/or venous thrombosis [12]. Splenomegaly and leukocytosis can also be found but is less frequently observed than it is in PV [13]. Diagnosis of ET is suggested when patients harbor a high platelet count. Bone marrow biopsy shows proliferative hematopoiesis mainly of the megakaryocyte lineage. More than 75% of mutations are associated with the targeting JAK2 transduction pathway ((JAK2V617F), calreticulin (*CALR*), or myeloproliferative leukemia virus (*MPL*); all these mutations induce constitutive JAK2-dependent signaling. ET patients do not suffer from a significant reduction in overall survival (OS) [14]; when effective treatments and management of the complications are prescribed, this pathology can lead to an OS of 20 years or more. However, in 10% of cases, ET patients will develop MF during evolution. Bone marrow proliferation, reticular fibers deposits, anemia, and age are the risk factors for poor ET evolution. In addition, AML transformation can also occur and the outcome of these secondary AMLs is poor. According to Europe Leukemia Net (ELN) recommendations, the goal of treatment for patients with PV and ET is to prevent first/recurrent thrombosis and bleeding while effectively controlling symptoms (e.g., hematocrit, splenomegaly, histologic changes, etc.) to minimize the risk of developing MF or AML. 

Primary myelofibrosis is characterized by erythroid-megakaryocytic hyperplasia in the bone marrow, neo-angiogenesis, and osteosclerosis. The median age of PMF at diagnosis is over 60 years; only 17% of patients are younger than 50. There is a male bias just as in PV (and not in ET) [15,16]. In PMF patients, common findings consist of anemia, fever, cachexia, bone pain, pruritus, splenomegaly, neutrophilia, and thrombocytosis. Immature granulocytes, nucleated red cells, and teardrop-shaped red cells in the blood are biological characteristics of this disorder. The bone marrow histology demonstrates magakaryocytosis with reticulin and/or collagen fibrosis grade 2 or 3 [17]. After relatively few years, 20% of PMF progresses to AML. Patients who did not transform may also succumb to other complications, including cardiovascular disease and the effects of cytopenias such as bleeding or inflammation [18]. PMF pathogenic mutations are in JAK2V617F in 50–60% of patients [19]. PMF therapy aims to improve survival and treat symptoms by preventing bone marrow failure and transformation to AML, treating common systemic symptoms, and preventing thrombohemorrhagic complications [20].

### 1.2. JAK-STAT Pathway

Janus kinase (JAK) is a non-receptor type of tyrosine protein kinase. The JAKs family consists of four isoforms containing JAK1, JAK2, JAK3, and TYK2. JAK2, as a member of the JAKs family, is highly homologous to the other members and is widely distributed in the cytoplasm of somatic cells. JAK2 is involved in signal transduction in the hematopoietic and immune systems and plays an essential role in the production of erythrocytes and activation of immune cells as well. Each of JAKs molecules comprises seven homologous domains (JH1-7) (Figure 1A), the carboxy-terminal JH1 (Jak-homology 1) is a catalytic site of the enzyme, whereas the amino terminal end (JH4-JH7) is known as the FERM domain (band 4.1, ezrin, radixin, moesin) and it is responsible for the binding of the kinase to the receptor. The JH3-JH4 domains of JAKs share high homology with the Src-homology-2 (SH2) domain, also known as the SH2 domain. JH2 (Jak-homology 2) shares high homology with JH1 but lacks enzymatic activity and is known as the pseudokinase region [21,22].

The STAT (signal transducer and activator of transcription) family contains seven subunits, including STAT1, STAT2, STAT3, STAT4, STAT5a, STAT5b, and STAT6. The activated JAKs can initiate specific STAT family members and interact with one another to constitute a complex regulatory network. The JAK-STAT system is one of the most important signaling pathways and involves three major components: (1) receptors, (2) JAK kinases, and (3) STAT [23,24]. The JAK2-STAT pathway, which is vital for communicating chemical signals from the outside of cells to the nucleus, is involved in cell division, cell death, tumor formation, and immunity [25]. 

In general, the cytokine receptor induces a conformational change upon ligand binding, resulting in the activation of receptor-associated JAK molecules and the phosphorylation of specific tyrosine residues within the intracellular domain of the cognate receptor. These phosphotyrosine residues on the receptor’s intracellular domain with an src homology-2 (SH2) or a phosphotyrosine binding domain can serve as docking sites for downstream signaling proteins. Once these messengers bind to the receptor, they are phosphorylated by JAK kinase, leading to its activation (Figure 2). In this way, extracellular signals can be delivered to activate several downstream effector processes, including STAT transcription factors, the RAS/MAPK pathway, and the PI3K/AKT pathway.

Abnormal cytokine signaling plays a vital role in the pathogenesis of MPN. In 1951, a study reported that MPN was caused by some unknown stimulus leading to excessive proliferation of BM cells [1]. Further studies revealed that hematopoietic progenitor cells from MPN patients exhibit hypersensitivity to cytokines such as erythropoietin (EPO), insulin-like growth factor-1 (IGF-1), IL-3, and granulocyte-macrophage colony-stimulating factor (GM-CSF), and were able to form erythroid and megakaryocyte colonies despite reduced or even absent cytokine levels (Figure 2) [26]. 

## 2. The Role of JAK2V617F Contributes to MPN: Basis in Structural Biology

### 2.1. The Potential Mechanism of JAK2 Auto-Inhibition

Currently, the tyrosine kinase of JAK2 does not have a complete structure and mainly consists of four fragment structures (JH1, JH2, SH2, and FERM) and three links (JH1–JH2 linker, JH2–SH2 linker, and SH2–FERM linker) connecting them (Figure 1). It is believed that the tyrosine kinase activity of JAK2 is primarily performed by the tyrosine kinase domain (JH1), perhaps involving a pseudokinase structural domain (JH2) as well, majorly phosphates the Ser523 and Try570. However, the detailed mechanism is not clear [27,28,29,30]. JAK2 kinase normal activity is activated by JH1 catalyzing two tyrosines (Try1007 and Try1008) situated on the active loop, phosphorylated Try1007 and Try1008 stabilizing the active state [31]. Hence, the trans-phosphorylation of these two tyrosines is a critical step for JAK activation [32]. Moreover, electrostatic interactions between JH1 and JH2 inhibit the tyrosine kinase activity of JH1 [33]. JH2 maintains the tyrosine kinase activity of JAK2 at a low basal level and maintains normal cellular activity; accordingly, it also indicated that JH2 has a tumor suppressor function [34].

The electrostatic interaction between JH2-JH1 is via two essential amino acids, phosphorylated in JH2, including Ser523 and Try570 [31,33]. The phosphorylation of Ser523 and Try570 is deemed to be an auto-phosphorylation of JH2 [35]. It has been also demonstrated that Ser523 is constitutively phosphorylated, whereas basal phosphorylation of Tyr570 is lower but increases significantly in response to cytokine stimulation in cells [27,28], suggesting that JH1 may also be involved in the phosphorylation of Try570. Phosphorylation of Try570 by JH1 was also critical in the JH2-JH1 in vitro assay [36]. These results indicated Ser523 may be exclusively auto-phosphorylated through JH2, whereas phosphorylation of Try570 may be achieved through the combined phosphorylation of JH1 and JH2. In addition, recent molecular dynamics models of JH2-JH1 have identified that pSer523 and pTry570 play an essential role in stabilizing the JH2-JH1 interaction as well (Figure 3) [31,37,38]. Moreover, other residues, including Met600, Lys603, Leu604, Lys607, Arg683, Glu592, and Arg588 in JH1 and Lys883, Lys926, Arg922, Asp873, Try931, Pro933, and Arg947 in JH2, are also involved in stabilizing the JH2-JH1 interaction through salt bridging or van der Waals forces [31,38]. Mutations in these amino acids impair JH2-JH1 interactions and JH1 phosphorylation activity. Nevertheless, Ser523 and Try570 can only participate in the JH2-JH1 interaction in the phosphorylated state, suggesting the pivotal role of Ser523 and Try570 in regulating JAK2 kinase activity.

Stabilization of JH2-JH1 can affect the flexibility of JH1 and JH2 lobes, which in turn would suppress the phosphokinase activity of JH1 due to the shift of JH1; and JH2 lobes have an instrumental role in the transfer of phosphate groups during protein phosphorylation [39]. Simultaneously, JH1 and JH2 interaction disrupts the Lys882 and Glu898 interaction in JH1, resulting in “DFG flip” promotion in the active loop [40,41], which affects the catalytic activity of JH1. In a molecular dynamics model, the idea that the interaction of JH1 and JH2 produces a “DFG out” state (which is a JH1 inactivation state) was established [31]. The three other JAKs have no reported catalytic activity for JH2, possibly due to the absence of a phosphorylation site. Ser523 and Tyr570 are not conserved in JAKs [35], suggesting that JAK2 may be the only JAK kinase that possesses this negative regulatory mechanism.

In addition, a study with insect cells identified that the JH2-JH1 construct from JAK2 is not fully auto-inhibited [36], suggesting the interaction between JH2-JH1 and FERM-SH2 is a component of full-length JAK2 protein auto-inhibition, and it appears that the interaction between FERM-JH2 and JH2-JH1 enhances JAK2 auto-inhibition. Recently, an interesting model of the active dimer was proposed based on the crystal dimerization of the FERM-SH2 of JAK2 bound to the erythropoietin receptor (EPOR) peptide [42], indicating that receptor-mediated formation of JAK2 FERM dimers is required for kinase activation with a separation of 120 Å or more between C termini and receptor transmembrane helices. A full-length JAK2 model indicates that the separation between receptor transmembrane helices in the active dimer is ~40 Å and ~90 Å in the inactive dimer [37], which is consistent with the other EPOR dimers [43,44]. Despite that, these findings implied that FERM is an integral part of JAK2 kinase activity.

### 2.2. The Potential Mechanism of JAK2V617F Hyper-Activation

The V617 mutation is not situated directly in the JH2-JH1 interface, but it is positioned close to the SH2-JH2 linker [31] and located at the FERM-JH2 interface as well [37]. The SH2-JH2 linker is essential for maintaining the basal state of JAK2 [45], and the FERM-JH2 interaction enhances the inhibition of JH1 phosphokinase activity via JH2. Therefore, the V617F mutation may disrupt the auto-inhibition of SH2 on SH1 by affecting the FERM-JH2, SH2-JH2, and SH1-JH2 interactions. It is also proposed that the V617F mutation not only affects the auto-inhibition of SH2 on SH1 but also stabilizes the active conformation of JAK2 through a novel molecular mechanism [38].

Initially, it was assumed that Ser523 and Try570 were auto-phosphorylated by JH2 either in cis or trans [30,43]. Moreover, the high-resolution crystal structure of JH2 and biochemical data revealed that Phe617 rotates the phenyl ring of Phe595 (αC in JH2) and induces a significant shift in the side chain position of the adjacent Phe594 [30]. The αC residue is situated at the JH2-JH1 interface and along the line of the ATP binding pocket (Figure 1B), a region with importance in cytokine-induced kinase activation [44,46]. Phe617, Phe595, and Phe594 formalized π-stacking interactions, and perturbation of Phe594 slightly alters the side chain position of Lys581 [43], which is a conserved lysine and forms a salt bridge with conserved glutamine in αC of JH2 [30]. A study identified the positions of Leu624 and Gln626, which are residues in the so-called “gatekeeper” position at the back of the ATP binding site, in the crystal structures of different ligands that appear to be compatible with the wild-type conformation of the JH2, but not with the V617F mutant conformation [45], due to steric clashes with Phe595 and Phe594, suggesting Phe617 and Phe595 interactions. Comparison of several available wild-types and V617F JH2 structures determined from various crystals reveal a distinct, stereotyped conformation of the β3-αC (JH2) loop and α-helix (αC) in the N lobe of JH2 (Figure 1B), where the main chain of the V617F mutant is shifted up to 6 Å relative to the wild-type structures [45]. This may explain why the catalytic activity of JH2 is impaired in V617F and JH2 loses its catalytic activity and would be unable to phosphorylate Ser523 and Try570, thereby disrupting the SH2-JH1 interaction. Although it was later found that JH1 was also involved in the phosphorylation of Try570, the hypo-phosphorylated Ser523 and Try570 still reduced the stability of SH2-JH1.

Subsequently, molecular dynamics simulations were performed, including JH2-JH1 and full-length JAK2 kinases. The JH2-JH1 simulations revealed that the bulky phenylalanine at residue 617 destabilized the position of the SH2-JH2 linker between JH2 and JH1 and, as a result, the catalytically active conformation of αC in JH1 was more stable in V617F than in wild-type JAK2. Also, in the simulations of double mutant V617F and F595A (located in the αC of JH2), which inhibits V617F activity [45,47], the SH2-JH2 linker position is again stabilized between JH2 and JH1 [31]. In addition, the model based on JH2-JH1 auto-inhibitory interactions [31] suggests that V617 is close to the SH2-JH2 linker and affects the conformation of the SH2-JH2 linker. In the model, the SH2-JH2 linker is wedged between JH2 and JH1, and the SH2-JH2 linker is shown to play a role in maintaining the JAK2 basal activity. The full-length JAK2 kinase simulations results revealed that V617F is located at the center of the FERM-JH2 interface in both the active and inactive monomer models, indicating that the abnormally high constitutive activity of the V617F mutant may not arise as the V617F mutation both weakens the inactive conformation and stabilizes the active conformation of JAK2 [37].

Recently it has been shown that the wild-type JAK2 and JAK2V617F activation networks are distinct. A possible mechanism for the activation of JAK2 by V617F is also proposed based on biochemistry and crystals. Activation of JAK2V617F is thought to be triggered by mutant V617F and delivered via two switch residues E596 and F537. These activation events are two distinct molecular mechanisms that, when combined, will result in the vigorous activity of this mutant. The first mechanism is the disruption of the inhibitory interaction between JH2 and JH1 (pY570-K883). The second mechanism is the positive regulatory interaction of K857 (JH1 N-term) mediated by F537 (SH2-JH2 linker C-term) [38].

## 3. The Role of JAK2V617F in Contributing to the MPN Phenotype

The JAK2V617F mutation is detected in the majority of MPN patients, and it is also an indicator of MPN in clinical diagnostics. However, it is common to hear the question asked in the literature why the same JAK2V617F is present in patients with related but distinct MPN. More than fifteen years from the original description of JAK2V617F, attributing the MPN phenotype to JAK2 over-activation only seems to be an oversimplification of the complex molecular interactions regulating the JAK/STAT pathway. Several studies have found that JAK2V617F does not provide a proliferative advantage to the HSC compartment [48,49,50,51]. Xenotransplantation of BM cells from JAK2 mutant patients into recipient mice showed that JAK2 mutations did not benefit from self-renewal. In contrast, JAK2V617F-positive cells did not enhance self-renewal but expanded at the progenitor cell level [52,53]. These results suggest that the JAK2V617F mutation is not sufficient to initiate MPN disease in the absence of other factors involved. Although JAK2V617F is causal with MPN, the development of MPN is a complex process based on a combination of differences in individual and cellular backgrounds. On the basis of the current studies, JAK2V617F mutant allele burden, different signaling pathways, epigenetics modification, the immune system, and lifestyle may all be involved in JAK2V617F associated MPN development (see Table 1 and Figure 4).

**Table 1 ijms-23-01013-t001:** JAK2V617F cooperate with different factors contribute to phenotype of MPN.

	Factors	Comments
JAK2V617F allele burden	- ET: homozygous - PV, ET: homozygous [54] - homo = 118 (104 PV, 14 ET) heter = 587, WT = 257	- Increased cardiovascular events (*p* = 0.013) - More frequent evolution into secondary myelofibrosis
	ET: allele burden > 50%. *n* = 165 [55]	Higher frequency of arterial thrombosis and splenomegaly
	PV: allele burden > 50%. *n* = 43 [56]	More severe disease status (higher HCT, RBC, HGB, and WBC)
	MPN: homozygous. heter = 45 homo = 13 [57]	Higher hemoglobin, increased incidence of pruritus, higher rate of fibrotic transformation
Distinct signaling	RAS-ERK and phosphatidylinositol 3 kinase-AKT pathways	Dysregulated erythropoiesis in PV
	IRS2 [58]	Increased cell viability and reduced apoptosis in JAK2-mutated cells
	IGF1R inhibitory [59]	Prevent the hematological disease in Jak2V617F mutation mice
	NT157 inhibits IRS1/2 and STAT3/5 [60]	
	- STAT3 pathway - STAT5, Ras/MEK/ERK and PI3K/Akt [61]	- Enhance LAP expression - Stimulates cell proliferation
Epigenetic modifiers	TET2, ASXL1, IDH1, IDH2, IKZF1 and EZH2	MPN-associated mutations
	TET2 12%, ASXL1 5%, DNMT3a 5%, EZH2 ~3% and IDH1 ~1.5% [62]	MPN
	- DNMT3A and TET2 - Mutation order of JAK2V617F and DNMT3A [63]	- Advantage to HSC/progenitor cells - Differences in MPN phenotype
	- ET: ~4% - PV: 9.8–16% - PMF:8–14% [64,65]	TET2 in different phenotype of MPN
Immune response	TNFa, IFNα, and IFNg pathways [66]	MPN development
	IFNα [67]	JAK2V617F increase molecular responses to
	TNFα [68]	Promote expansion of JAK2V617F cells in MPN
	ROS and inflammatory factors [69]	influence MPN progression
Effect of lifestyle	Smoking [70]	Increase the risk of MPN
	Coffee consumption [71]	Inversely associated with the risk of PV
	Mediterranean dietary [72]	Decrease symptom burden in MPN
	Obesity [73]	Elevate overall risk for MPN especially with ET

### 3.1. JAK2V617F Mutation Allele Burden

Clinical data suggest that JAK2V617F mutations are present in 95% of patients with PV and 50–60% of patients with ET and PMF [13,74,75,76], with homozygous JAK2V617F accounting for approximately 25–30% of patients with PV and 2–4% of patients with ET [5,54,75]. Vannucchi et al. found that JAK2V617F homozygosity could identify symptomatic myeloproliferative disease in patients with PV or ET. This study was based on 118 homologous JAK2V617F patients (including 104 PV, 14 ET), 587 JAK2V617F heterozygous patients, and 257 wild-type patients [54]. Antonioli et al. reported that the JAK2V617F mutant allele burden contributes to determining the clinical phenotype of patients in thrombocythemia [55]. Consistent with this, Kim et al. showed that the JAK2V617F mutant allele burden was positively associated with HCT, RBC, HGB, and WBC in patients with PV, and the mutant allele burden more significant than 50% reflects a severe disease state [77]. In addition, one hypothesis proposes that the ratio of mutant to wild-type JAK2 is critical to the phenotype, and this hypothesis has been validated by mouse models [52,57]. Tiedt et al. revealed that the mice developed an ET-like phenotype when JAK2V617F expression was lower than endogenous wild-type JAK2; the mice developed a PV-like phenotype with thrombocytosis and neutrophilic features when JAK2V617F levels were approximately equal to wild type JAK2, and higher levels of JAK2V617F caused a PV-like phenotype but no platelet formation [6]. The expression of homozygous JAK2V617F in the granulocyte compartment is much more common in PV than in ET [3]. In PV, JAK2V61F homozygous is correlated with the elevated HCT, WBC, splenomegaly, and lower platelets [78]. This hypothesis proposed that homozygous JAK2V617F drives the erythroid phenotype. A murine study showed that a low level of JAK2V617F drives thrombocytosis and a slight increase in HCT, which is the phenotype resembling ET, whereas higher level of JAK2V617F cause erythrocytosis and leukocytosis without thrombocytosis [57].These results suggest that lower levels of JAK2V617F are predominantly present in ET, whereas higher levels are present in PV.

### 3.2. Distinct Signaling in JAK2V617F MPN Phenotype

It has been indicated that the homodimeric type I cytokine receptor is necessary for optimal signaling by JAK2V617F [79,80] and several signaling cascades activated by mutation JAK2V617F, including STAT5, MAPK, and PI3K pathways [61,81,82]. At least some aspects of the STAT5 pathway have been documented as necessary and sufficient for the MPN phenotype, in both in vitro and in vivo experiments [83,84,85]. Analysis of gene expression from normal and JAK2V617F heterozygous samples revealed that STAT5 activation is common in both PV and ET, whereas STAT1 activation is significantly pronounced in ET patients compared with PV patients [86]. Increased STAT1 activity promotes megakaryopoiesis and constrains erythropoiesis of cord blood-derived CD34^+^ cells, which support a clue for the phenotypic differences of specific MPN [87]. 

Insulin receptor substrate (IRS) proteins have reported involvement in myeloid neoplasms [58,88], and they include adaptor proteins that could be activated by receptors participating in hematopoietic signaling, including IGF1R, EPOR, and TPOR [59,60,89]. Subsequent evidence indicates that IRS2 is associated with JAK2V617F induced malignant transformation and upregulation of IGF1R signaling, further inducing the MPN phenotype [60]. Reactive oxygen species (ROS) signaling also potentially contributes to the MPN phenotype [90]. It was shown in a mouse model that JAK2V617F induced ROS production, which would lead to genomic instability and disease occurrence [91]. In humans, ROS was increased in the hematopoietic cells of MPN patients compared to healthy individuals [92,93]. Smoking can further aggravate the impact of chronic inflammation and oxidative stress in MPNs, thereby contributing to genomic instability, subclone formation with MPN-disease progression, and the increased risk of second cancers [94]. High mobility group A2 (HMGA2, a non-histone architectural transcription factor) and let-7 (which negatively regulates HMGA2 expression) abnormal signaling activity can lead to a distinct clinical phenotype in MPN patients with JAK2V617F [95]. In addition, Uras et al. revealed that cyclin-dependent kinase 6 (CDK6) promotes MPN by enhancing cytokine production (in conjunction with NF-kB), activating leukemic stem cells, and preventing apoptosis [96]. Given that JAK2V617F is involved in multiple signaling regimes, most of the signaling mechanisms remain unclear.

### 3.3. Epigenetic Modifiers in JAK2V617F MPN Phenotype

Many MPN patients harbor somatic mutations in epigenetic regulators as well. For instance, DNA methylation (including TET2, DNMT3A, and IDH1/2) or chromatin structure (contains ASXL1 and EZH2) and accessory mutations in the regulators are proposed to affect MPN specific phenotypes [63,97,98], promote progression, and induce phenotypic switching. In addition to driver mutations, alterations in the function of epigenetic regulators may also act as disease modifiers of MPN [99]. In addition, the order in which the mutations occur is critical. TET2 mutations altered the transcriptional program of JAK2V617F activation in a cell-intrinsic manner and are more frequently seen in the ET phenotype. Conversely, patients who acquire JAK2V617F first are more likely to develop PV [100]. TET2, ASXL1, and EZH2 mutations, either prior or subsequent to the acquisition of JAK2V617F, have been documented to affect hematopoietic stem and progenitor cell biology and clinical performance [62,63]. Moreover, Chen et al. demonstrated that loss of TET2 driving the clonal advantage of HSCs and expression of JAK2V617F led to an expanded population of downstream progenitors, which led to disease progression through a combination of effects [101]. Staehle et al. reported that the Jumonji domain containing 1C (JMJD1C), a histone demethylase, and it is a nuclear factor erythroid 2 (NFE2) target gene, may affect JAK2V617F driven myeloproliferative disease in mice [102]. Additionally, Peeken et al. showed that overexpression of NFE2 in MPN is associated with JAK2V617F phosphorylation of histone H3Y41 [103]. These results revealed that DNA methylation plays a crucial role in JAK2V617F associated MPN progression and it is probably the best described MPN in epigenetics mechanisms. McKenney et al. reported that mutations in epigenetic regulators, including mutations in isocitrate dehydrogenase 1 (IDH1), IDH2, and TET2, were identified in patients who progressed from MPN to AML [104,105]. In addition, co-expression of mutant IDH1 or IDH2 with JAK2V617F enhanced the progression of MPN in mice [103].

### 3.4. Immune Response in JAK2V617F MPN Phenotype

There is still a debate about the immune response in MPN, but some evidence has shown that immune response is critical in the progression of MPN. When analyzing the expression of the inflammatory genes in MPN patients, JAK2V617F heterozygous clones expressed p-selectin significantly higher than wild-type cells in the same patient [66]. In addition, tumor necrosis factor-a (TNFa), interleukin 6 (IL6), and interferon gamma (IFNg) signaling were detected enriched in heterozygous JAKV617F clones compared to the cells from healthy individuals [67]. The absence of TNFa in JAK2V617F transduced bone marrow cells completely abrogates the MPN phenotype in a mouse retroviral bone marrow transplant model [68]. Jacquelin et al. reported that the absence of DNMT3A enhanced TNFa signaling and inflammation in the JAK2V617F retroviral transduction and transplantation model, suggesting that MPN progression is a complicated multi-signal interaction [69]. Moreover, the role of NFkB/CDK6 in the regulation of inflammatory cytokines was further highlighted in the JAK2V617F transgenic mouse model [96]. Transformation and clonal evolution in MPN and other cancers were linked to genomic instability, which could be triggered by inflammatory cytokines [106,107]. Moreover, ROS is associated with genomic instability, point mutations, and chromosomal aberrations in patients with JAK2V617F positive MPN [108,109], suggesting that ROS and inflammatory factors may cross interact to influence MPN progression. Holmström et al. demonstrated that a specific CD8^+^ cytotoxic T lymphocyte selectively recognized JAK2V617F mutant cells, suggesting that cancer immunotherapy is a potential new treatment modality for MPN [110].

### 3.5. Effect of Lifestyle in JAK2V617F MPN

The JAK2V617F mutation is one of the main acquired somatic mutations in MPN. However, the JAK2V617F mutation can be detected in the general population. Hinds et al. identified JAK2V617F carriers with a prevalence of 0.17% in the general population. The prevalence of JAK2V617F was associated with age and gender in this study [111]. It is well known that aging, smoking, and gender are the main risk factors for MPN. However, in addition to this, individual lifestyle also greatly influences the relationship between JAK2V617F and MPN. An NIH-AARP study found coffee consumption has an inverse relation to MPN risk where compared to low levels of caffeine intake. High intake of caffeine was also protective against risk for PV development [71]. The Mediterranean dietary pattern helps prevent common chronic diseases such as obesity and type 2 diabetes. A study also showed that the Mediterranean dietary help MPN patients decrease more than half of the symptom burden compared with MPN patients not following a Mediterranean diet [72]. It was also reported that the BMI was positively associated with the ET risk. Recently, the MOSAICC study showed that the obesity appeared to elevated overall risk for MPN, especially with ET [94]. These studies suggest that modification of lifestyle maybe a useful strategy to prevent the progression or transformation of MPN, especially for early stage MPN, and to reduce symptom burden.

## 4. Summary and Discussion

MPN is thought to arise in the HSC pool due to a single HSC acquiring a somatic mutation that alters the mutant HSC cell fate and promotes myeloid differentiation to produce a myeloproliferative phenotype. JAK2V617F is the most common mutation detected in the majority of patients with MPN and the mutant over-activates the JAK-STAT signaling pathway, which involves multiple signals mediating cell division and differentiation contributing to MPN progression. JAK2 inhibitors are also clinically effective against some types of MPN, suggesting the hyperactivity of JAK2 in MPN. Increasingly, studies demonstrated that JAK2V617F triggers MPN and requires the assistance of additional factors, for instance, different signaling pathway, epigenetic modification, and the immune response. The cross-linking of these signals forms a complex network of interactions and creates differences in individual and cellular contexts, determining the development of the MPN phenotype.

It is worth noting that studies revealed patients or ‘healthy individuals’ who are positive with JAK2V617F mutation who never develop MPN or other diseases [112,113] Recently, studies used state-of-the-art lineage tracing to determine the time of clonal amplification and disease progression after the acquisition of the JAK2V617F mutation [114,115] and suggested that the individuals who acquire the mutation in utero or early childhood could be diagnosed with disease decades later or even live with this mutation for life and without disease development. In addition, differences in individual immunity could likely have an influence, as previously described that specific CD8^+^ cytotoxic T lymphocytes selectively recognize JAK2V617F mutant cells [110]. In addition, a large number of homozygous JAK2V617F clones could be seen in male patients with PV, whereas female patients manifested ET, suggesting that gender may also regulate the phenotypic consequences of homozygosity [116]. Moreover, external factors may also be involved, for instance, a significantly higher risk of ET and unspecified MPN, but no PV was found in association with a history of smoking [117,118]. Furthermore, there were gender differences in the association of smoking history with MPN, as women with a history of smoking had a higher risk of developing MPN than men [66]. Importantly, it demonstrated that JAK2V617F positive ET and PV form a biological continuum [74], indicating that ET is an early stage of PV progression. Conversion to PV is rare in ET patients with the JAK2V617F mutation; nevertheless, it still occurs in some cases. A study showed that 5% of 422 JAK2V617F mutation ET patients converted to PV after 13 years [119]. The other study reported that there is 29% conversion rate of PV after 15 years in 466 ET patients with JAK2V617F mutation [120]. There is a study with mouse models showing that the phenotype of ET mouse models can convert to PV after several months of follow-up [121]. It was further demonstrated that the induction of MPN by JAKV617F requires the assistance of additional factors, including internal and external factors. In conclusion, other studies are necessary to investigate the contribution of JAK2V617F to the occurrence of MPN and provide more support for the treatment of MPN.

## Figures and Tables

**Figure 1 ijms-23-01013-f001:**
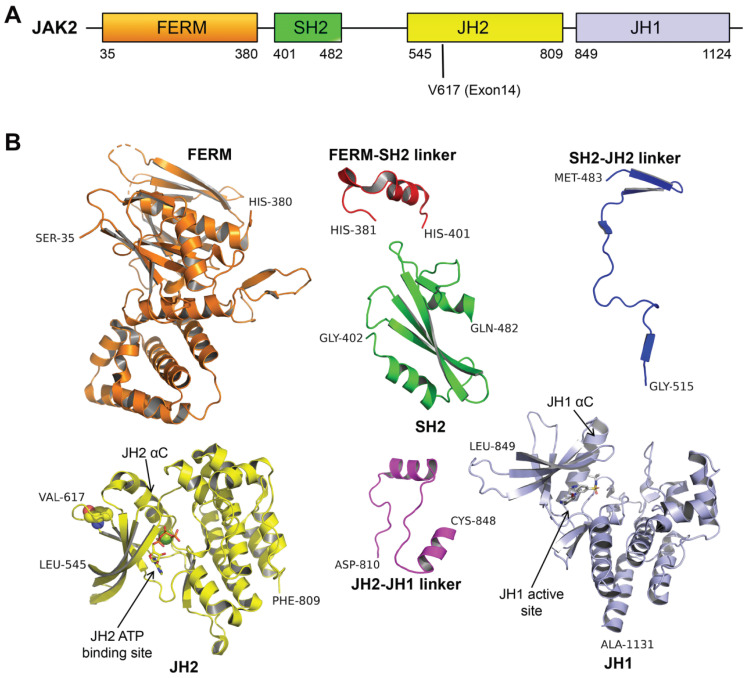
The structure of JAK2. (**A**) The domain organization of full length JAK2. (**B**) JAK2 domains and linkers were adapted from PDB and drawn in different colors. FERM (band 4.1, ezrin, radixin, moesin) domain colored in brown, SH2 (an Src homology 2) domain colored in green, JH2 (Jak-homology 2) domain colored in yellow, JH1 (Janus homology-1) domain colored in light blue, FERM-SH2 linker colored in red, SH2-JH2 linker colored in blue, and JH2-JH1 colored in magenta. FERM, FERM-SH2 linker, SH2, and SH2-JH2 linker (PDBID: 6E2Q); JH1 (PDBID: 6VNE); JH2 (PDBID: 5I4N); and JH1-JH2 linker (PDBID: 4OLI).

**Figure 2 ijms-23-01013-f002:**
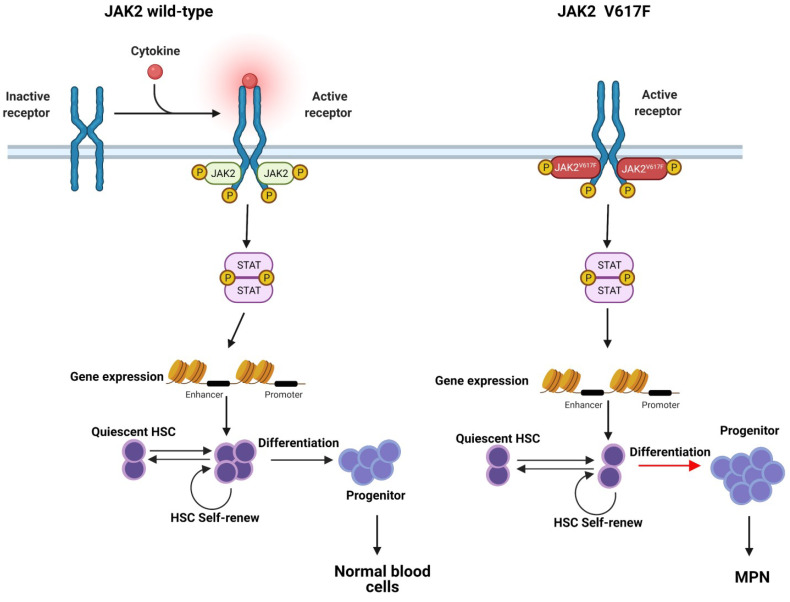
Typical and JAK2V617F induced dysregulated JAK2-STAT signaling. Normal JAK-STAT signaling is mediated by cytokines and growth factors, for instance, EPO and TPO, which maintain an ordered balance of cell proliferation and differentiation by self-renewal in HSC cell pool and are essential for normal blood cell formation (left panel). The JAK2V617F mutation causes JAK2 to be more sensitive to cytokines and consistently activates the JAK-STAT pathway. The over-activation of JAK2-STAT signaling alters the critical cell fate and leads to more progenitor cell development and further progression to MPN (right panel).

**Figure 3 ijms-23-01013-f003:**
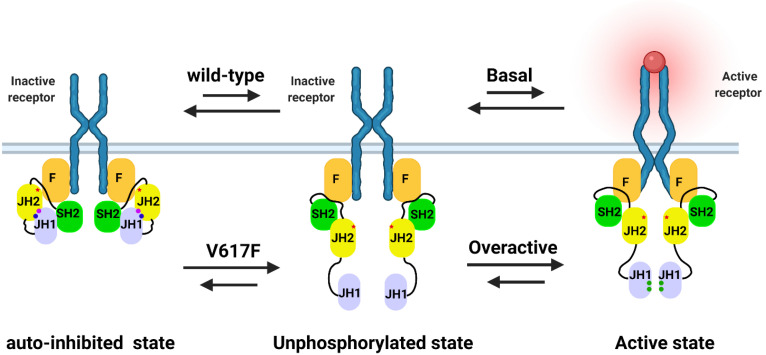
Model of the interactions between the domains of the JAK2 protein. The three main states of JAK2 include auto-inhibited state, unphosphorylated state, and active state. In the auto-inhibited state, the protein with phosphorylated S523 (pS523) and phosphorylated Y570 (pY570) are locked in the inactive conformation. The unphosphorylated state is a transition between the active and inactive conformations. In the active state, the two JAK2 molecules associated with a cytokine receptor dimer are maintained in positions that trans-phosphorylation of the JH1 activation loop, including Tyr1007 and Tyr1008. The pS523 is drawn as a blue dot, the pY570 is drawn as a red dot, the V617F mutation is drawn as a red star, and the pY1007/1008 is drawn as green dots.

**Figure 4 ijms-23-01013-f004:**
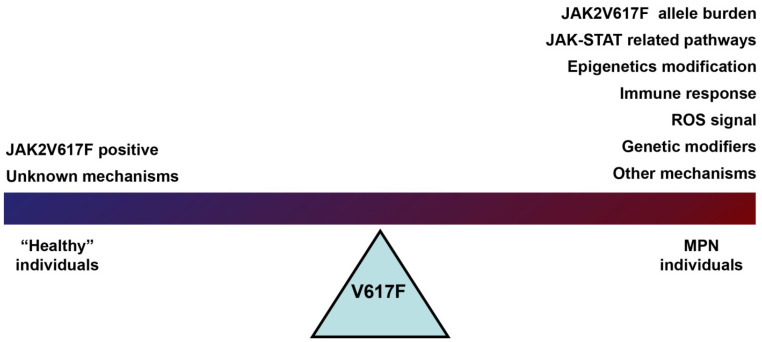
Model of JAK2V617F positive contributing to MPN phenotype.

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
