# Peer review of "Insights into the Potential Mechanisms of JAK2V617F Somatic Mutation Contributing Distinct Phenotypes in Myeloproliferative Neoplasms"

_ijms, 2022, doi:10.3390/ijms23031013_

Round 1

Reviewer 1 Report

To the authors,

Even though this work is a consolidated review of all findings in the area of MPN, you have mentioned at the beginning that "this current study JAK2V617F may underlie the development of MPN, with the assistance of
additional contributing factors such as JAK-STAT-related signaling pathways, epigenetic modifications, and immune responses".

You will need to rephrase this as review of the work in the related field.

2. Figure 1:

The cartoon that is shown to differentiate between JAK2 WT and V617F mutation is not very well depicted, please add the names of the cytokines and what does the red arrow indicate? 

3. What is the relevance of including life style and and V617F mutation and MPN?

Author Response

Point 1: Even though this work is a consolidated review of all findings in the area of MPN, you have mentioned at the beginning that "this current study JAK2V617F may underlie the development of MPN, with the assistance of additional contributing factors such as JAK-STAT-related signaling pathways, epigenetic modifications, and immune responses".

You will need to rephrase this as review of the work in the related field.

Response 1: It’s a great suggestion to modify this sentence to make our paper more professional. We’ve modified this sentence in the paper as “Here we review the major MPN phenotypes, JAK-STAT pathways and mechanisms of development based on structural biology, while also describing the impact of other contributing factors such as gene mutation allele burden, JAK-STAT-related signaling pathways, epigenetic modifications, immune responses and lifestyle on different MPN phenotypes.”

Point 2: The cartoon that is shown to differentiate between JAK2 WT and V617F mutation is not very well depicted, please add the names of the cytokines and what does the red arrow indicate?

Response 2: Thanks for this reminder, added the cytokines name makes the cartoon easier to read and understand. We’ve added the TPO, EPO, GM-CSF, IL3, IL5 into the cytokines of Figure1. We use the red arrow to more significantly distinguish of HSC differentiation of the JAK2V617F mutation differs from that of WT

Point 3: What is the relevance of including life style and and V617F mutation and MPN?

Response 3: A research showed that the V617F mutation expressed in majority of MPN patients but also in the general populations. And some studies also found the good lifestyle habits (like Mediterranean dietary pattern, low BMI) can prolong the onset of MPN, slow the progression of MPN and reduce the severity of MPN symptoms. In the “Effect of lifestyle in JAK2V617F MPN” of part three :The role of JAK2V617F in contributing to the MPN phenotype, we showed some studies  support this point.

Reviewer 2 Report

This study reviewed the potential mechanisms of JAK2V617F somatic mutation contributing distinct phenotypes in myeloproliferative neoplasms. The review is necessary and comprehensive. I have some comments to improve the manuscript:

- Presentation style and structure should be improved. For example, the abstract does not need to be separated into different paragraphs, there are many sub-sections that can be merged.

- The authors should describe clearly how did they pool the data/references.

- The authors should have some statistical analyses to show the differences among studies.

Author Response

Point 1: Presentation style and structure should be improved. For example, the abstract does not need to be separated into different paragraphs, there are many sub-sections that can be merged.

Response 1: That’s a good idea to improve the structure of the paper. The abstract was separated into two paragraph to state the JAK2V617F is not the major parts but have other factors influence the phenotype of MPN. But if merged abstract into one paragraph is also a great way to showed this paper as an abstract. So we merged the abstract in one paragraphs. Introduction part is classified into two parts, one is MPN which contain different 3 sub-types and another one is JAK-STAT pathway. If we separate the 3 sub-types in different sub-sections will makes some confuse for the reader,so we merged these 3 parts into one and separate introduction into two parts. And we merged the two paragraphs of each sub-types into one paragraph. The part two is mainly about structure biology of JAK2V617F in MPN. In this part we discussed the potential mechanism of JAK2 from auto-inhibition and hyper-activation in structure biology. The third part is mainly discussed the JAK2V617F role in MPN phenotype. This part was composed with 5 sub-sections to clearly showed the JAK2V617F in different phenotype to understand why the same gene mutation cause different phenotype. This part can merged into less sub-sections but will not clearly identify the potential mechanisms of different phenotype. And fourth part is make a conclusion for this review.

Point 2: The authors should describe clearly how did they pool the data/references.

Response 2: In this review our data were come from different papers as we used references. And in some sentence the references do not clearly showed in each, we modified them in the paper. Each data we discussed were also added the reference as well. The specific content of the figure is not clearly shown in the paper. We have made changes in the article to explain our content more clearly for these mistakes.

Point 3: The authors should have some statistical analyses to show the differences among studies.

Response 3: Show the differences among studies is a great way to show the analysis. We made a table to show the differences about the role of JAK2V617F in contributing to MPN from five different aspects. In this table we can clearly read the different factors influence JAK2V617F contribute to phenotype of MPN.